# The Role of Physical Activity-Related Health Competence and Leisure-Time Physical Activity for Physical Health and Metabolic Syndrome: A Structural Equation Modeling Approach for German Office Workers

**DOI:** 10.3390/ijerph181910153

**Published:** 2021-09-27

**Authors:** Simon Blaschke, Johannes Carl, Jan Ellinger, Ulrich Birner, Filip Mess

**Affiliations:** 1Department of Sport and Health Sciences, Technical University of Munich, 80992 Munich, Germany; jan.ellinger@tum.de (J.E.); filip.mess@tum.de (F.M.); 2Department of Sport Science and Sport, Friedrich-Alexander University Erlangen-Nürnberg, 91058 Erlangen, Germany; johannes.carl@fau.de; 3Siemens AG, Human Resources EHS, Department of Psychosocial Health and Well-Being, Otto-Hahn-Ring 6, 81739 Munich, Germany; ulrich@birner.net

**Keywords:** health literacy, physical literacy, physical health, metabolic syndrome, office workers, health management, structural equation model

## Abstract

Office workers (OWs) are prone to insufficient physical activity (PA), which increases their risk of metabolic syndrome (MetS) and impaired physical health. The Physical Activity-related Health Competence (PAHCO) model holds the potential to facilitate a healthy physically active lifestyle. Therefore, in this study, we investigate the interplay between PAHCO, leisure-time PA, physical health, and MetS in OWs in Germany. In a cross-sectional study, OWs (*N* = 316, 25% female) completed self-report questionnaires along with an occupational health checkup to examine their Metabolic Syndrome Severity Score (MetSSS) values. Structural equation modeling indicated a strong positive association between PAHCO and leisure-time PA and a small positive association with physical health. PAHCO showed a considerable negative association with the MetSSS. Leisure-time PA was a positive mediator for the PAHCO–physical health association but was not a significant mediator for the association between PAHCO and the MetSSS. These findings underscore the importance of PAHCO in the context of leisure-time PA, physical health, and MetS in OWs. Furthermore, our findings highlight the health-enhancing value of the qualitative aspects of PA, such as motivational and volitional components in PA participation, with respect to physical health and MetS.

## 1. Introduction

Noncommunicable diseases (NCDs) such as cardiovascular diseases, diabetes, and cancer cause ~70% of deaths worldwide and ~90% of deaths in Germany [1,2]. Metabolic syndrome (MetS) is an important indicator [3] for identifying people at risk of NCDs in alignment with the World Health Organization’s (WHO’s) strategy for prevention. MetS concerns the co-occurrence of several cardiovascular risk factors (e.g., insulin resistance, abdominal obesity, and hypertension), demonstrating good predictive quality for NCD incidence [4,5]. In addition, the WHO has targeted health behaviors, specifically physical activity (PA), as the best strategy for promoting health and mitigating NCDs’ burden [6].

Research has shown that PA can substantially improve physical and metabolic health [7,8]. PA’s positive effects on health are highlighted by the term “health-enhancing physical activity” (HEPA), which encompasses all forms of PA that benefit health without causing undue harm or risk [9]. Despite the health benefits that can be achieved via PA [10], about one-quarter of adults worldwide do not meet the WHO recommendations to exert or perform at least 150 min of moderate-intensity or 75 min of vigorous-intensity PA per week [11]. The need for PA promotion is also evident in German working adults, as questionnaire-based studies have suggested that ~47–53% do not fulfill the WHO recommendations for sufficient PA [12,13].

Office workers (OWs), who predominantly operate in desk-based and digitally assisted occupational activities, including remote work [14], represent more than one-third of the overall workforce in Germany [15]. Owing to prolonged periods of sedentary activities and insufficient PA, OWs are at risk of developing musculoskeletal disorders (MSDs) and MetS [16,17,18]. Research on MSDs and MetS has indicated a reciprocal connection between these risk factors [19]. The overall proportion of OWs in the German workforce, the increased prevalence of MSD and MetS health risks, and the interconnectedness of these risks in OWs highlight the particular relevance of HEPA promotion with this target group.

Findings by Holzgerve et al. [20] have supported a positive influence of HEPA interventions in OWs on work-related MSDs and the closely related construct of physical health. In addition, Ryu et al. [21] have underscored the value of HEPA in the prevention of MetS in an intervention study designed to address OWs’ health. However, research on PA and OWs’ health is scarce [22]. Furthermore, studies without a specific focus on OWs have highlighted the potential for leisure-time PA to improve metabolic health by compensating for high occupational sitting time and insufficient overall PA behavior simultaneously [23,24]. Although leisure-time PA has shown a positive influence on health [25], occupational PA often displays an inverse association with health, which can, for instance, be due to low individual control of PA in the workplace [26]. These findings have underlined the importance of research on HEPA in OWs to ease the occupational health risks derived from long periods of sitting.

### 1.1. Introducing Physical Activity-Related Health Competence

Although the demand for HEPA promotion is apparent, particularly in OWs, explaining and understanding PA behavior is more complex [27], and personal factors provide the largest explanatory evidence [28]. Therefore, personal factors are embedded in the “Global Action Plan on Physical Activity 2018–2030” of the WHO, which highlights the importance of physical and health literacies and competencies to increase HEPA [29]. Literacy [30,31] and competence [32] addressing PA or health show considerable conceptual overlap and can guide human behavior [33].

The latest approaches related to the promotion of HEPA underscore the importance of integrating physical, motivational, and cognitive dimensions [33]. However, most constructs do not fully exploit the potential of personal factors for HEPA because health and PA do not represent similarly meaningful interactive components [34,35].

The Physical Activity-related Health Competence (PAHCO) model is located at the interface between physical literacy and health literacy [36,37]. The authors of this model have assumed that PAHCO benefits individuals’ health and well-being by promoting HEPA and aims to tailor PA toward leading a healthy physically active lifestyle [38]. PAHCO specifies three subcompetencies to facilitate HEPA (Figure 1, right) [33].

First, movement competence enables individuals to master essential locomotor tasks in daily situations (e.g., climbing stairs or lifting objects) and to participate in planned exercise sessions during leisure time (e.g., running or cycling) [35]. The second subcompetence, control competence, consists of the two facets of control competence for physical training and PA-specific affect regulation. Control competence for physical training involves individuals’ knowledge of physical training to model load and intensity for ensuring efficient gains for physical health [39]. Next to this, PA-specific affect regulation aims to produce positive psychological health changes through PA. The third subcompetence of the PAHCO model, PA-specific self-regulation competence, consists of personal dispositions and motivational–volitional characteristics, which serve as the psychological basis for regular HEPA [36].

The integration of each subcompetence into action relies on a set of interacting basic elements that serve as a prerequisite for transferring competence to HEPA behavior (Figure 1, left) [37,40]. The basic elements of movement competence consist of motor abilities and motor skills combined with body awareness (e.g., endurance, strength, coordination, and balance). Control competence characterizes the accurate knowledge of the consequences of PA behavior (“effect knowledge”) and the situational application of PA and exercise with respect to the HEPA prerequisites (“action knowledge”) [36]. The basis of control competence consists of selecting, perceiving, and processing body signals caused by PA to model beneficial outcomes for physical and mental health [35]. The underlying basic elements of PA-specific self-regulation consist of positive attitudes toward HEPA and high PA-specific tasks and behavior self-efficacy. In addition, the authors of the PAHCO model have highlighted the interactions among the basic elements and subcompetencies to facilitate HEPA [33].

### 1.2. Current Research on the Role of PAHCO in PA and Health

Recent research on the PAHCO model underpins the structural validity of the subcompetencies and analyzes the interplay of the concept with PA and health [33,37]. Several studies have confirmed the positive association between PAHCO and the amount of PA in adults, ranging from rehabilitation patients and university sport participants to apprentices in nursing and automotive mechatronics [36,41]. On the subscale level, PA-specific self-regulation displays the largest positive link to PA [36,42].

Sudeck et al. [38] have found that control competence moderates the link between PA and affective well-being. PAHCO has also displayed a positive direct and indirect connection to physical health indicators [36]. Similar to these findings, apprentices in nursing and automotive mechatronics have shown a positive link between PAHCO and psychophysical health [41]. In addition, Carl et al. [41] have questioned the direct link between the amount of PA and psychophysical health, because these two constructs exhibit only an indirect positive relationship mediated by PAHCO. This finding highlights the need for a more detailed focus on the functional–qualitative aspects of PA that qualify for HEPA [43].

As of yet, no study has addressed the relationships between the PAHCO and occupational and leisure-time PA separately. Research on the interplay of PAHCO and health-related outcomes has not targeted OWs specifically. Lastly, Schmid et al. [42] have underlined the importance of including objective health parameters, as multiple studies have reported a positive link between the PAHCO, PA, and health, exclusively on the basis of self-report measures.

### 1.3. Research Goals and Hypotheses

These theoretical considerations highlight the need to conduct research concerning PA behavior in OWs for the promotion of physical and metabolic health. So far, the PAHCO model has not been used specifically with OWs. Increasing the knowledge on the direct and indirect associations of PAHCO in the context of PA and health in OWs is important because of the occupational risk factors affecting their health [44]. Above all, this is the first study to include perceived physical health as a subjective indicator for physical health and MetS as an objective indicator for metabolic health in analyses of the connections between PA and PAHCO. Consequently, we describe the interplay between PAHCO, leisure-time PA, and physical and metabolic health, resulting in the following hypotheses:PAHCO has a direct positive relationship with leisure-time PA in OWs.PAHCO has a direct positive relationship with physical and metabolic health in OWs.PAHCO has an indirect relationship with physical and metabolic health mediated by leisure-time PA in OWs.

## 2. Methods

### 2.1. Participants

Of the 446 employees eligible during the data collection period, 388 (87%) gave written informed consent to participate in the study. A total of 327 (73%) employees met the inclusion criterion of being employed in an office work occupation and having the absence of acute illness or disease prohibiting HEPA. We excluded four employees as univariate outliers on the leisure-time PA index and also disregarded the data of one employee as a univariate outlier on the component score of the mental health scale and the MetS. In addition, five people were detected as multivariate outliers. Thus, 316 (71%) OWs remained in the analyses. The final sample consisted of 236 (75%) men and 80 (25%) women, with a mean age of 50.9 years (standard deviation (*SD*), 6.4 years). Table 1 displays the participants’ sociodemographic parameters by gender.

### 2.2. Materials

The study consisted of self-report questionnaires assessing (1) PAHCO, (2) occupational PA, (3) leisure-time PA, and (4) mental and physical health. In addition, the participants took part in an occupational health checkup to assess their MetS parameters.

#### 2.2.1. Self-Report Measures

Participants answered sociodemographic questions concerning gender, age, relationship status, education attained, and medication use. The questionnaires in this study are widely used measures and demonstrate good reliability and validity in the German working-age population [36,45,46,47].

(1)The Questionnaire on the PAHCO model developed by Sudeck and Pfeifer [36] contains the three subscales of control competence for physical training, PA-specific affect regulation, and PA-specific self-regulation with a total of 13 items and responses on a 4-point Likert scale. Control competence for physical training (e.g., “If I want to enhance my health by strengthening my trunk muscles (back, stomach), I am confident that I know the right exercises to do”) consists of six items (Cronbach’s α = 0.87). PA-specific affect regulation (e.g., “If I am feeling down, I can distract myself well through physical activity”) is assessed with four items (Cronbach’s α = 0.88). PA-specific self-regulation (e.g., “When I decide to do more exercise, I am very disciplined in implementing this plan.”) comprises three items (Cronbach’s α = 0.84). The PAHCO questionnaire comprises mean scores on the three subscales and a total mean score of all 13 items ranging from 1 to 4. Higher mean scores resemble better PAHCO for the total mean value and the subscales (Cronbach’s α = 0.88).The subcompetence of movement competence is not part of the questionnaire because, at the time of data collection, no scale on PAHCO existed that included movement competence [39]. The absence of movement competence in this measure of PAHCO may also be due to the difficulty of universally valid assessment tools for motor competence measured across different populations [48].(2)Participants completed the work dimension of the Baecke Physical Activity Questionnaire [49] with eight items (e.g., “What is your main occupation?”) to strictly include OWs in this study and assess the sample’s PA level during work. The questionnaire examines occupational PA with a weighted mean score ranging from 1 to 5, with low values indicating low occupational PA.(3)The Godin–Shepard Leisure-Time Physical Activity Questionnaire (GSLTPAQ) [50] has six items that determine individuals’ PA during leisure time at mild, moderate, and vigorous intensity (e.g., “Over the last 7 days (i.e., the last week), how many times on average did you do the following kinds of exercise for more than 30 min during your free time?”). The GSLTPAQ results in the cumulative weighted leisure score index, which displays the amount of leisure-time PA, with values >24 indicating sufficient leisure-time PA [51].(4)Physical and mental health was operationalized by the Short-Form Health Survey (SF-12) version 2.0 [47,52], which examines self-reported health with a weighted and standardized component score on physical (Cronbach’s α = 0.79) and mental (Cronbach’s α = 0.86) dimensions, with six items each. The mental component score and the physical component score range from 0–100, with a mean of 50 and a standard deviation of 10. High values indicate better self-reported health.

#### 2.2.2. Objective Measures

In addition to the self-report measures, this study examined the metabolic syndrome severity score (MetSSS), which was assessed in a voluntary occupational health checkup. The MetSSS consists of the parameters also used in the MetS classification of the National Cholesterol Education Program Adult Treatment Panel-III (NCEP ATP III) [53], which are blood pressure, waist circumference, triglycerides, high-density lipoprotein-cholesterol, and fasting glucose [54]. Participants were instructed to refrain from eating and drinking for eight hours before their checkup. The blood pressure of the participants was assessed after a five-minute rest in a recumbent position with calibrated manual blood pressure cuffs (Clinicus II; Bosch + Sohn GmbH & Co. KG, Jungingen, Germany). At the end of expiration, the participants’ waist circumference was examined manually with a tape measure in an upright standing position at the center between the lowest rib and the highest prominent point of the iliac crest. Blood serum was collected in venipuncture from the antecubital vein (S-Monovette, Sarstedt AG & Co. KG, Nümbrecht, Germany). Triglyceride, high-density lipoprotein-cholesterol, and fasting glucose levels were obtained in laboratory analyses (SYNLAB Holding Deutschland GmbH, Augsburg, Germany).

The MetSSS is a centered and scaled indicator of metabolic health, with a mean of 0 and *SD* of 1 by weighting blood pressure, waist circumference, triglycerides, high-density lipoprotein-cholesterol, and fasting glucose by gender and ethnicity [54]. Lower MetSSS values indicate better metabolic health. The MetSSS allows a precise quantification of metabolic risk and provides additional prediction of future diabetes and NCDs beyond the MetS classification of the NCEP ATP III [55]. This study used the formula for non-Hispanic White men and women, which is by far the largest ethnic group of working adults in Germany, to calculate the MetSSS [56]. The MetSSS displayed excellent validity compared to the MetS classification by the NCEP ATP III in this study (area under the curve = 0.94).

### 2.3. Procedures

This study originated from a project to evaluate the occupational health checkup and the comprehensive stationary workplace health promotion programs (WHPPs) of a large, global, private-sector company in Germany. As part of this project, this study comprises quantitative, cross-sectional, monocentric data collection from April 2019 to December 2019, combining the participants’ self-report measures and the MetSSS of the participants at the start of a three-week comprehensive WHPP.

The company’s employees were informed via email or brochure regarding the study’s aim and scope. All participants provided their informed written consent before taking part in the paper-and-pencil survey and completing the occupational health checkup. An external physician who was independent from the project performed the checkup to examine the employees’ MetS. This study fulfilled the company’s data privacy guidelines, was in accordance with the Declaration of Helsinki, and received approval from the Ethics Committee of the School of Medicine of the Technical University of Munich (IRB number: 645/20 S-KH).

### 2.4. Data Analysis

We impute the missing data values by applying multivariate-chained equations [57]. We excluded the univariate and multivariate participant outliers from the analysis following the recommendations of Tabachnick and Fidell [58]. Means (*M*), standard deviations (*SD*), and bivariate correlations (*r*) were calculated for the total PAHCO score and its subscales, occupational and leisure-time PA, the physical component score, the mental component score, and the MetSSS. The significance level for the bivariate correlations was set to *p* < 0.05.

The assumptions of linearity, normality, homoscedasticity, and independence of the residuals were met prior to the multivariate regression analyses and the structural equation modeling (SEM) [59]. The standardized estimates (β) of the multivariate linear regressions guided the inclusion of the sociodemographic parameters of gender, age, relationship status, education, and medication use for the subsequent SEM. We adjusted the significance level of the multivariate regression analyses to *p* < 0.0125 by applying Bonferroni correction due to possible multiple testing issues [60].

The SEM included the three PAHCO subscales as latent first-order factors, explaining the respective items of the PAHCO questionnaire as manifest factors. Moreover, the three first-order factors were explained by one latent second-order factor representing PAHCO. On the basis of previous research on PAHCO, the latent second-order factor was a direct predictor of leisure-time PA and the correlated criterion variables of the physical component score and the MetSSS [35,36]. In addition, leisure-time PA directly predicted the physical component score and the MetSSS. Because of the primary focus on the criterion variables of physical health and the MetSSS and the results of the bivariate correlations, the mental component score served as a control variable for the physical component score. Accordingly, occupational PA was solely included as a control variable for the MetSSS.

We assessed the global model fit for the SEM using the chi-square/df value (χ2/df: acceptable, ≤4; good, ≤2), the comparative-fit index (CFI: acceptable, ≥0.95; good, ≥0.97), the root-mean-square error of approximation (RMSEA: acceptable, ≤0.08; good, ≤0.05), and the standardized-root-mean-square residual (SRMR: acceptable, ≤0.10; good, ≤0.05) [61]. In addition to the fit indices, standardized loadings (λ) served to examine the structural validity of the PAHCO model and provided the basis for testing the study’s hypotheses. The significance level of the SEM was set to *p* < 0.05.

In addition, we calculated the 95% confidence intervals (CIs) and standard errors (*SEs*) of the standardized estimates (β) and the adjusted proportion of explained variance by the predictors (*R*^2^) for the SEM. All effect sizes in the analyses were standardized and interpreted as small (≈0.10), moderate (≈0.30), or strong effects (≈0.50) [62]. The data preparation and statistical analyses were conducted using R and RStudio (Version 3.4.3; RStudio Inc., Boston, MA, USA) [63].

## 3. Results

### 3.1. Bivariate Correlations and Multivariate Linear Regressions

The total PAHCO (*r* = 0.42, *p* < 0.001) score displayed a moderate positive association with leisure-time PA. The correlation of the total PAHCO score with the physical component score showed a moderate positive effect size (*r* = 0.26, *p* < 0.001). The total PAHCO score indicated a moderate negative effect size to the MetSSS (*r* = −0.21, *p* < 0.001). For the three PAHCO subscales, PA-specific self-regulation showed the strongest relationships with leisure-time PA (*r* = 0.45, *p* < 0.001), the physical component score (*r* = 0.26, *p* < 0.001), and the MetSSS (*r* = −0.22, *p* = 0.01). The association of leisure-time PA to the physical component score (*r* = 0.28, *p* < 0.001) was positive, with a moderate effect size. In addition, leisure-time PA showed a small negative correlation to the MetSSS (*r* = −0.13, *p* = 0.02).

Next to these associations with direct reference for the hypotheses, other bivariate correlations and the multivariate regression analyses guided the subsequent SEM. For instance, the physical component score correlated negatively with the MetSSS (*r* = −0.18, *p* = 0.001). Occupational PA displayed a small positive effect size regarding the connection to the MetSSS (*r* = 0.14, *p* = 0.01). Table 2 illustrates all bivariate correlations of this study.

The multivariate linear regression analyses examined higher scores of the total PAHCO mean score for older participants (β = 0.13, *p* = 0.02) and participants with no medication intake (β = 0.29, *p* = 0.01). Male participants displayed higher MetSSS in comparison to female participants, with a strong effect size (β = 1.02, *p* < 0.001). A moderate negative effect size was found for no medication use on the MetSSS (β = −0.42, *p* < 0.001). No medication use displayed a positive moderate effect size on leisure-time PA (β = 0.32, *p* = 0.006) and the physical component score (β = 0.40, *p* < 0.001). The complete results of the multivariate linear regression are displayed in Appendix A (Table A1).

### 3.2. SEM of the Second-Order PAHCO Factor

The global fit indices of the SEM were good to acceptable (χ^2^(169) = 232.41, χ^2^/df = 1.38, CFI = 0.96, RMSEA = 0.04, SRMR = 0.05). The first-order factors of control competence for physical training (0.62 ≤ λ ≤ 0.82; *p* < 0.001), PA-specific affect regulation (0.66 ≤ λ ≤ 0.90; *p* < 0.001), and PA-specific self-regulation (0.75 ≤ λ ≤ 0.84; *p* < 0.001) showed strong positive standardized loadings on the 13 items of the PAHCO questionnaire. The second-order PAHCO factor loaded significantly on the three first-order factors with strong-to-moderate effect sizes (0.48 ≤ λ ≤ 0.92; *p* < 0.001).

On the grounds of the overall model fit and the structural validity of the PAHCO model, the main purpose of the SEM was to answer the hypotheses of this study (Table 3, Figure 2). The association of the second-order PAHCO factor to leisure-time PA was strong, with a positive effect size (β = 0.52, *p* < 0.001). The model explained 29% of the adjusted variance for leisure-time PA. The second-order PAHCO factor showed a small positive effect size for the path to the physical component score (β = 0.16, *p* = 0.02) and a small negative path to the MetSSS (β = −0.22, *p* < 0.01). The association between leisure-time PA and the physical component score was positive, with a small effect size (β = 0.14, *p* = 0.01). The path between leisure-time PA and the MetSSS was not significant in the SEM (β = −0.01, *p* = 0.90). The predictor variables explained 27% of the adjusted variance for the physical component score and 26% of the adjusted variance for the MetSSS.

In addition to these primary outcomes, PAHCO was positively related to small effect sizes to the participants’ age (β = 0.15, *p* = 0.02) and participants reporting no intake of medication (β = 0.19, *p* < 0.01). The mental component score displayed a strong positive effect size with the physical component score (β = 0.40, *p* < 0.001). Participants stating no intake of medication showed higher physical component scores with a small effect size compared to participants reporting medication intake (β = 0.10, *p* = 0.04). Participants with no medication intake also had lower MetSSS values (β = −0.15, *p* = 0.001). Men had higher MetSSS values than women (β = 0.41, *p* < 0.001). The complete SEM is presented in Appendix B (Figure A1).

## 4. Discussion

The goal of the study was to examine the direct relationship between PAHCO and (1) leisure-time PA and (2) physical and metabolic health. Furthermore, this study aimed to investigate (3) the indirect relationship between PAHCO and physical health and the MetS mediated by leisure-time PA.

### 4.1. Direct Relationship between PAHCO and Leisure-Time PA

The first hypothesis assumed a positive direct relationship between PAHCO and leisure-time PA. Our results confirm this hypothesis by indicating a positive relationship between PAHCO and leisure-time PA. In addition, the three PAHCO subscales were positively related to leisure-time PA, with PA-specific self-regulation showing the strongest effect size and PA-specific affect regulation displaying the smallest positive effect size.

In general, our findings support those of previous research on the relationship between PAHCO and the self-report measures of leisure-time and overall PA [36,41]. However, this study showed a stronger effect size for the association between PAHCO and leisure-time PA as compared to that calculated by prior research [41]. The stronger effect size of PAHCO to leisure-time PA could result from factors such as low individual control in occupational PA, which is subsumed under overall PA and thereby impairs the association between PAHCO and overall PA [26]. Our findings support this assumption, with PAHCO displaying no linear relationship with occupational PA. Furthermore, Sudeck and Pfeifer [36] have corroborated this assumption by demonstrating moderate-to-strong positive relationships between PAHCO and the leisure-time PA facet of habitual sport activities for participants in university sports.

The results of our study highlight the importance of distinguishing between different facets of PA for the effect size of the relationship between PAHCO and PA [16] and the relevance of PAHCO for leisure-time PA in OWs. Sudeck and Pfeifer [36] have found the same pattern for the connections of the three PAHCO subscales to leisure-time PA. PA-specific self-regulation and control competence for physical training displayed strong, respectively moderate, positive connections with leisure-time PA. The value of these two subscales for leisure-time PA can be explained in alignment with the social–cognitive theory of PA [28]. Control competence for physical training comprises PA-related knowledge, which helps build the intention to engage in leisure-time PA with respect to the social–cognitive theory of PA [64,65]. PA-specific self-regulation consisting of PA-related self-efficacy can strengthen the motivational and volitional determinants for leisure-time PA [36]. Self-efficacy presents the pivotal construct of the social–cognitive theory of PA by enabling a lasting enhancement in PA behavior [66].

In addition to this theory-driven explanation of our findings, our results underscore the importance of including sociodemographic factors in the overall analyses. Participants who stated no intake of medication and the participants’ age displayed a positive small connection to PAHCO. While these factors have not been addressed explicitly in the research concerning PAHCO, studies on health literacy and age report conflicting results for working aged adults, but agree on the overall influence of sociodemographic parameters on health literacy [67].

In summary, our findings on the positive connection between PAHCO and leisure-time PA extend the results of prior research on OWs. In accordance with the social–cognitive theory of PA, PAHCO might increase PA-related knowledge and improve the individual motivational and volitional determinants of PA behavior. These changes on the personal level can enable leisure-time PA gains in OWs, which can reduce the occupational health risk of physical inactivity in this target group. Conversely, PAHCO can increase through experience-based learning during participation in leisure-time PA [33,68].

### 4.2. Direct Relationship between PAHCO and Physical and Metabolic Health

Our second hypothesis assumed a positive direct relationship between PAHCO and physical and metabolic health. In this study, we found a positive direct relationship with a small effect size between PAHCO and physical health and a negative moderate direct relationship with the MetSSS. These findings support our second hypothesis and indicate a potentially health-enhancing connection between PAHCO and physical and metabolic health in OWs.

For the link between PAHCO and physical health, the results of our study follow those of other recent research, which displayed positive connections between PAHCO and outcomes related to physical health [36,39,41]. However, the comparability of these results to our findings is limited because the other studies did not focus on OWs and used different measures related to physical health, yet the majority of previous research has examined slightly larger effects for this association.

The differences in the connection between PAHCO and physical health could be due to work-specific factors, such as occupational sitting time [16] and mental work-related demands [69], which account for larger shares of the physical health variance in OWs compared to other occupational groups, such as blue-collar employees who perform physical and manual work (e.g., manufacturing and landscaping) [70]. In addition, Besharati et al. [71] have underscored the relevance of mental work-related demands and occupational sitting time in OWs’ physical health. The predominance of these occupational factors could explain the smaller effect size for the connection between PAHCO and physical health in OWs.

Although the effect size for the PAHCO–physical health connection varied, our study supported a direct, positive connection between both. This connection might arise because PAHCO can guide individuals’ perceptions of physical health and the respective adaption of the physical load during leisure-time PA to enable physical health benefits.

We also investigated the association between PAHCO and MetS. Our findings generally conform to those of prior research on health literacy, which have determined that higher health literacy levels are related to a lower risk of MetS [72,73,74]. These findings support the value of PAHCO and health literacy in the prevention of MetS and NCDs. In addition to the health-enhancing value of PAHCO and health literacy for people at risk of MetS, another commonality of our results with previous research concerns the vital role of sociodemographic variables, with male participants displaying a strong effect size in relation to a higher risk profile of MetS in comparison to female participants [73,75]. Next to these commonalities, methodological differences between these studies on health literacy and our research on PAHCO can impede the quantitative comparison of the effect sizes for the association of health literacy and PAHCO with MetS.

Despite these differences, Yokokawa et al. [73] have highlighted the link between the health literacy aspect of decision-making based on health-related information and MetS. This is in line with our findings examining the strongest link between PAHCO and MetSSS for the subscale of PA-specific self-regulation. PA-specific self-regulation, with a detailed focus on motivational and volitional characteristics, shares substantial common ground with the health literacy aspect of decision-making, as volitional factors in particular can facilitate the behavioral enactment of intentions [76,77].

Our study transferred the positive connection between PAHCO and the physical health of OWs. In addition, we extended the results of the connection between health literacy and MetS to the domain-specific PAHCO model. The results on the direct connection between PAHCO and physical health and MetS provide additional insights into the value of PAHCO in relation to these subjective and objective health outcomes, with particular relevance for OWs.

### 4.3. Mediating Role of Leisure-Time PA on the Relationship between PAHCO and Physical and Metabolic Health

Our third hypothesis assumed a mediating role for leisure-time PA in the relationship between PAHCO and physical and metabolic health. Leisure-time PA was a small positive mediator in the relationship between PAHCO and physical health but displayed no mediating role for the relationship between PAHCO and the MetSSS. These results confirm the third hypothesis for physical health while rejecting the assumed indirect link between PAHCO and the MetSSS mediated by leisure-time PA.

Prior research has delivered consistent results for the mediating effect of facets of leisure-time PA on the relationship between PAHCO and physical functions [36]. Our analysis also showed a mediating role of leisure-time PA on the connection between PAHCO and physical health in OWs. These results underline the theoretical background of the PAHCO model [39], which postulates that competencies shape and promote PA to improve health. In OWs, the mediating effect of leisure-time PA on the link between PAHCO and physical health can mitigate the occupational health risk of increased sedentariness, which often results in MSDs and impaired physical health [20].

The direct relationship between PAHCO and physical health examined in our mediation analysis indicated partial mediation, suggesting the influence of additional theoretical mechanisms [78]. This partial mediation is in agreement with findings by Sudeck et al. [38], which examined the moderating effect of PAHCO on the link between PA and well-being, implying a potential reciprocal mechanism between PAHCO and leisure-time PA. This assumption is supported by the closely related constructs of health and physical literacies by displaying a reciprocal connection between health behavior and literacy [31,68]. These results and the general theoretical background can inform future considerations of the PAHCO model [36], which currently suggests an unidirectional link between PAHCO and HEPA.

Regarding the mediating role of leisure-time PA on the link between PAHCO and the MetSSS, to the best of our knowledge, no other study has addressed PAHCO and objective health outcomes. Our findings, however, are not in line with the results of the review by Zhang et al. [79], who have examined a direct negative relationship of leisure-time PA with MetS. Although the results of the bivariate correlation between leisure-time PA and the MetSSS assessed in our study agreed with the review, none of the reviewed studies included PAHCO to address the link between leisure-time PA and MetS. These methodological and conceptual differences can explain the deviation of our results.

The results from our mediation analysis indicated that PAHCO might hold the potential to shape the health-enhancing qualitative aspects of leisure-time PA and prevent MetS. Similarly, Carl et al. [41] found no mediating effects of PA volume on the connection between PAHCO and psychophysical health. These findings, along with our results on the mediating role of the connection between PAHCO and MetS in OWs, underscore the importance of the qualitative aspects of PA having a positive effect on health.

Holtermann et al. [26] have supported a focus on the qualitative aspects of PA by stating that extreme forms of leisure-time PA might result in overtraining and have an impairing relationship with cardiovascular health parameters. Following this notion, leisure-time PA and HEPA might have differing effects on metabolic health outcomes, particularly at the individual level. Pardo et al. [80] have shared this understanding, as HEPA can also occur while commuting to the workplace or during household tasks.

The present study adopted the suggestion of Carl et al. [39] by including leisure-time PA and physical health as sequential outcomes of PAHCO and supporting the theoretical background of this model in relation to physical health. In addition, the partial mediation of the connection between PAHCO and physical health and the nonsignificant mediating relationship of leisure-time PA on the direct connection of PAHCO and MetSSS point toward a reciprocal promotion of health outcomes by the interconnectedness of PAHCO and leisure-time PA. These results imply the importance of qualitative aspects of leisure-time PA to promote health.

### 4.4. Strengths and Limitations

To the best of our knowledge, ours is the first study to investigate the relationships between PAHCO and leisure-time PA, physical health, and MetS, which particularly addresses OWs. Focusing on OWs and PAHCO might be crucial because OWs are exposed to the occupational health risks of physical inactivity, impaired physical health, and MetS, and PAHCO has the potential to promote these health-related outcomes [21].

Another strength of our study is the SEM approach with the inclusion of a latent second-order PAHCO factor and the manifest factors of leisure-time PA, physical health, and MetS [61]. This statistical approach allowed the replication of the structural validity of the PAHCO model and the assessment of the direct and indirect relationships between PAHCO, leisure-time PA, physical health, and MetS. Thus, the SEM approach can foster future considerations for the conceptual advancement of the PAHCO model.

In addition, this study enriches the current state of research on PAHCO by including the objective health outcome of MetS in the analysis, as no other study on PAHCO has, as of yet, included objective health outcomes in the analysis [42]. Although this approach avoids the issue of common method variance, the connection between PAHCO and MetS highlights the value of PAHCO in relation to MetS, which is an important indicator of numerous NCDs.

However, our study also has some limitations that future research should consider. We did not record movement competence in this study. At the time of data collection, movement competence was not incorporated into the PAHCO questionnaire owing to the difficulty of operationalizing this subcompetence across varying physical levels in healthy and clinical adult samples and to the questionable validity of self-reporting movement competence [36]. However, the current version of the PAHCO questionnaire has resolved this issue by validating self-reported movement competence across various adult samples [39]. This subcompetence could provide additional insights into the relationships between PAHCO and health, with movement competence comprising locomotor abilities and skills, which are key requirements for HEPA participation and, consequentially, for physical health and MetS.

Furthermore, future research on PAHCO in OWs should reconsider the assessment of PA. This reconsideration could imply measuring self-reported HEPA, as our findings indicated partial to no mediation of leisure-time PA. HEPA is the conceptual outcome of the PAHCO model, and leisure-time PA and HEPA have shown conceptual differences [80].

Moreover, subsequent studies on PAHCO and MetS in OWs could profit from the inclusion of objective PA measurements using accelerometers, with previous research having suggested moderate to no agreement between the subjective and objective measures of PA [81,82]. The accelerometer-based PA assessment can allow for the examination of sedentariness. Sedentariness is substantially distinct from physical inactivity and presents a crucial parameter in the context of OWs’ health [16,23], which might also play a vital role in relation to PAHCO and MetS.

Lastly, the SEM approach of our study presented insight into the direct and indirect relationships between PAHCO and leisure-time PA, physical health, and MetS, but did not convey causality on the role of PAHCO in the context of these relationships over the course of time. Longitudinal and interventional studies could address the reciprocal nature of the relationship between PAHCO and leisure-time PA or address the meaning of PAHCO for the maintenance of sufficient levels of leisure-time PA in relation to physical health PAHCO and MetS.

## 5. Conclusions

The results of our study contribute to existing research on PAHCO and OWs’ health by testing the PAHCO model in the context of the MetSSS as an objective health outcome, introducing the PAHCO model with respect to the relationships between leisure-time PA, physical health, and MetS. This study examined health-promoting relationships between PAHCO and leisure-time PA, physical health, and the MetSSS. In addition, the mediation analysis of leisure-time PA on the relationship between PAHCO and the MetSSS particularly suggested the value of qualitative aspect of leisure-time PA to mitigate MetS, as opposed to solely focusing on the amount of PA. In summary, with OWs being prone to the occupational health risks of physical inactivity, MSDs, and MetS, our findings point towards the relevance of PAHCO for WHPPs in this target group. On the one hand, implementing the PAHCO model in WHPPs in OWs might hold the potential to increase OWs’ leisure-time PA, promoting physical and metabolic health. On the other hand, WHPP employing PAHCO could also focus on sedentariness during work and infer health-promoting effects by changing the work-related behavior of OWs. The motivational and volitional characteristics of PA-specific self-regulation could, for example enhance the frequency of active breaks or walking meetings in this target group. Next to the potential practical value of PA-specific self-regulation, control competence for physical training might enable OWs to develop PA behavior, which incorporates the occupational risk factors and meets their individual demands to promote physical and mental health. Therefore, future longitudinal and interventional studies on PAHCO should place a holistic perspective on PA, sedentariness, and physical and metabolic health and critically review our findings on PAHCO in OWs to substantiate them, with the aim of creating WHPPs on PAHCO in this target group.

## Figures and Tables

**Figure 1 ijerph-18-10153-f001:**
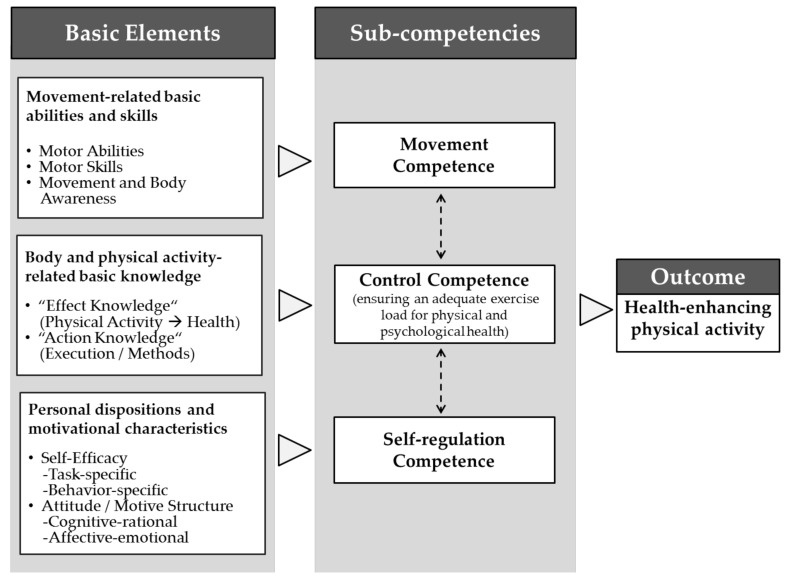
The Physical Activity-related Health Competence (PAHCO) model [36].

**Figure 2 ijerph-18-10153-f002:**
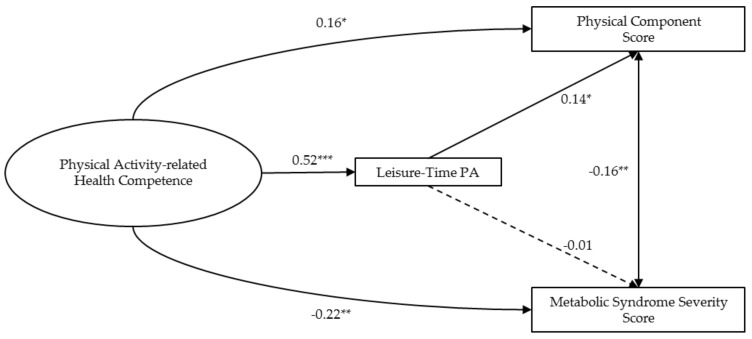
Standardized path coefficients for the predictor, mediator, and criterion variables. Notes. * *p* < 0.05, ** *p* < 0.01, *** *p* < 0.001; For simplification, first-order factors, manifest variables of the PAHCO Questionnaire, and covariates are not reported in the figure.

**Table 1 ijerph-18-10153-t001:** Participants’ sociodemographic characteristics.

	Total (*N* = 316)	Men (*n* = 236)	Women (*n* = 80)
Age, *M* (*SD*)	50.9 (6.4)	51.4 (6.4)	49.5 (6.2)
Relationship status			
Relationship, *n* (%)	256 (81.0)	195 (82.6)	61 (76.3)
Single, *n* (%)	60 (19.0)	41 (17.4)	19 (23.7)
Education			
Tertiary, *n* (%)	203 (64.2)	172 (72.9)	31 (38.7)
Secondary, *n* (%)	105 (33.2)	60 (25.4)	45 (56.3)
Primary, *n* (%)	8 (2.5)	4 (1.7)	4 (5.0)
Medication intake			
No medication, *n* (%)	142 (44.9)	109 (46.2)	33 (41.2)
Medication, *n* (%)	174 (55.1)	127 (53.8)	47 (58.8)

Notes. *N* = total sample; *n* = sub-sample; *M =* mean; *SD* = standard deviation; tertiary = college degree; secondary = vocational training; primary = high school qualification.

**Table 2 ijerph-18-10153-t002:** Means, SDs, and bivariate correlation coefficients for the main variables.

	1.	2.	3.	4.	5.	6.	7.	8.	9.
1. PAHCO									
2. CCPT	**0.84 *****								
3. PAAR	**0.70 *****	**0.30 *****							
4. PASR	**0.74 *****	**0.47 *****	**0.38 *****						
5. Occupational PA	0.03	0.01	**0**.04	**0**.04					
6. Leisure-time PA	**0.42 *****	**0.34 *****	**0.20 *****	**0.45 *****	0.03				
7. Physical Component Score	**0.26 *****	**0.23 *****	0.11	**0.26 *****	−0.01	**0.28 *****			
8. Mental Component Score	**0.12** *	**0.16 ****	−0.00	**0.11 ***	0.11	0.10	**0.45 *****		
9. MetSSS	**−0.21 *****	**−0.12 ***	**−0.17 *****	**−0.22 ***	**0.14 ***	**−0.13 ***	**−0.18 ****	−0.01	
*Mean*	2.68	2.64	2.78	2.60	1.66	18.42	49.01	45.97	0.06
*SD*	0.48	0.58	0.64	0.65	0.28	14.27	6.18	7.94	0.86

Notes. *N* = 316, * *p* < 0.05, ** *p* < 0.01 (two-tailed), *** *p* < 0.001 (two-tailed); PAHCO = Physical Activity-related Health Competence; CCPT = Control Competence for Physical Training; PAAR = Physical Activity-Specific Affect Regulation; PASR = Physical Activity-Specific Self-Regulation; Significant bivariate correlations are displayed bold.

**Table 3 ijerph-18-10153-t003:** Standardized path coefficients, SEs, and 95% CIs for the predictors.

Predictor	β	*SE*	95% CI
Mediator: Leisure-time PA			
PAHCO	**0.52 *****	**0.06**	**(0.41, 0.62)**
Medication intake	0.06	0.05	(−0.01, 0.17)
	*R*^2^ = 0.29		
Criterion: Physical Component Score			
Leisure-time PA	**0.14 ***	**0.05**	**(0.03, 0.25)**
PAHCO	**0.16 ***	**0.07**	**(0.04, 0.29)**
MetSSS	**−0.16 ****	**0.06**	**(−0.28, −0.04)**
Mental Component Score	**0.40 *****	**0.05**	**(0.32, 0.49)**
Medication intake	**0.10 ***	**0.05**	**(0.00, 0.20)**
	*R*^2^ = 0.27		
Criterion: MetSSS			
Leisure-time PA	−0.01	0.06	(−0.13, 0.11)
PAHCO	**−0.22 ****	**0.08**	**(−0.37, −0.08)**
Physical Component Score	**−0.16 ****	**0.06**	**(−0.28, −0.04)**
Medication intake	**−0.15 ****	**0.05**	**(−0.24, −0.06)**
Sex	**0.41 *****	**0.04**	**(0.32, 0.49)**
Occupational PA	0.07	0.05	(−0.03, 0.17)
	*R*^2^ = 0.26		

Notes. *N* = 316, * *p* < 0.05, ** *p* < 0.01, *** *p* < 0.001 (two-tailed); PAHCO = Physical Activity-related Health Competence; MetSSS = Metabolic Syndrome Severity Score; β = standardized coefficients; *SE* = standard error; 95% CI = 95% confidence interval; *R*^2^ = adjusted proportion of explained variance; medication intake = reference group is participants using medication; sex = reference group is female; Significant regression paths are displayed bold.

## Data Availability

The datasets generated and analyzed during the study are not publicly available owing to patient confidentiality but are available in a highly anonymized form from the corresponding author on reasonable request.

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
