# Peer review of "The Role of Physical Activity-Related Health Competence and Leisure-Time Physical Activity for Physical Health and Metabolic Syndrome: A Structural Equation Modeling Approach for German Office Workers"

_ijerph, 2021, doi:10.3390/ijerph181910153_

Round 1
Reviewer 1 Report
This article is ok for publication.
Author Response
Dear Reviewer 1,
I appreciate the time and effort that you dedicated to providing feedback on our manuscript and are grateful for the insightful comments on and valuable improvements to our paper. I sincerely hope the revised manuscript reflects most of your suggestions.
Kind regards,
Simon Blaschke

Reviewer 2 Report
The work presents a clear structure and purpose, with a good methodological structure.
The use of the structural equation model is an interesting perspective for the analysis of the relationships that the authors propose, however the statement of the hypotheses seems somewhat generic since it was very likely that these favorable relationships would exist. Perhaps it would have been interesting to specify the degree of relationship since the SEM model can show the degrees of relationship between the variables.
Author Response
Dear Reviewer 2,
Thank you for the time and the effort that you dedicated to providing feedback on our manuscript. We are grateful for the insightful comment on our paper.
We thoroughly considered incorporating the suggestion on the specification of the degree in the hypotheses and hope our response to your comment in the attachments makes the decision made comprehensible.
Kind regards,
Simon Blaschke

Reviewer 3 Report
The authors of the study undertook a very difficult task of analyzing the risk factors of the metabolic syndrome related to physical activity in people with a mostly sedentary lifestyle related to work. They analyzed the Physical Activity–related Health Competence (PAHCO) model, and its relationship with physical and metabolic health mediated by leisure-time PA. The work is methodologically correct, interesting and valuable. I believe it should be published.
Author Response
Dear Reviewer 3,
We appreciate the time and effort that you dedicated to providing feedback on our manuscript and are grateful for the insightful comments and the acknowledgement of the quality of our research.
Kind regards,
Simon Blaschke

Reviewer 4 Report
Thank you for the thoughtful and important study. As mentioned in the paper, this study is the first of its kind to look into the relationships of PAHCO with leisure-time PA, physical health, and MetS of office workers. The significance of the findings is widely applicable, not only within Germany, but to the global urban society. I appreciate the clear explanation of the findings, and the reflection at the end of the paper about further refinement of the research design. This study not only addresses important life conditions current in our world, but also opens up new directions for further and larger scale studies.
Author Response
Dear Reviewer 4,
Thank you for the time and the effort that you dedicated to providing feedback on our manuscript. We are grateful for the insightful comments and appreciate the acknowledgement of the scientific vigor our paper.
Kind regards,
Simon Blaschke

Reviewer 5 Report
I have carefully read the manuscript titled “The Role of Physical Activity–related Health Competence and Leisure-Time Physical Activity for Physical Health and Metabolic Syndrome: A Structural Equation Modeling Approach for German Office Workers” (ijerph-1368894).
The manuscript investigates the association of the Physical Activity-related Health Competence (PAHCO) with respect to physical activity in leisure time, physical health and the risk for Metabolic Syndrome in a consistent group of office workers.
I have found the manuscript interesting, well written and organized.
Only some considerations from my side. Although I appreciated the accurate analyses performed by the Authors on the subjective and objective parameters investigated, from an occupational health perspective, it could be interesting to more deeply define/discuss how the use of such model in workplace settings may support health promotion plans and improve physical activity, thus balancing the risks related to the sedentary work they perform. In this view longitudinal studies should be important also to see the effectiveness of such programs on the PAHCO, leisure physical activity, as well as physical health of employees.
Some considerations on possible differences due to age and level of education should be included in the text. These aspects are extremely important as potential influencing factors in the analyses.
I suggest the Authors to better explain the meaning of “blue collar employees” for the readers that are not familiar with this definition.
Finally in the phrase: “In addition to these primary outcomes, participants stating no intake of medication showed higher physical component scores with a small effect size compared to participants reporting no medication intake (β = 0.10, p = 0.04)”, there seems to be something not correct.
Author Response
Dear Reviewer 5,
We are grateful for the insightful comments on our paper. We have been able to incorporate changes to reflect most of the suggestions provided by your revision and have highlighted the changes within the manuscript and phrased a detailed response to your comments in the attachments.
Kind regards,
Simon Blaschke

Reviewer 6 Report
This study investigated the interplay among PAHCO, leisure-time PA, 18 physical health, and MetS in OWs in Germany.
For this study, OWs completed self-report questionnaires along with an occupational health checkup to examine 20 their Metabolic Syndrome Severity Score (MetSSS) values.
Generally, the importance of this investigation was recognized and well deserved.
The abstract covers strictly the main statements of the publication, Introduction is short but concise,
Methods are adequate for the purposes. (It needs IRB number).
Especially, this manuscript explains well about the Introducing Physical Activity–related Health Competence, Physical Activity–related Health Competence (PAHCO) model, and even Current Research on the Role of PAHCO in PA and Health.
However, the narration of introduction dealing with the specific mechanism about the necessity of this research with much more references.
It should be much better if this study tested about movement competence.
It needs to be described about specific rationale (the correation betwen PAHCO and metabolic syndrome) and references in the discussion section.
Author Response
Dear Reviewer 6,
We appreciate the time and effort that you have dedicated to providing your valuable feedback on my manuscript. We are grateful to the reviewer for their insightful comments on my paper. We have been able to incorporate changes to reflect most of the suggestions provided by you. We have highlighted the changes within the manuscript and attached a response to your comments.
Kind regards,
Simon Blaschke
